# IMPROVED FULLY QUANTIZED TRAINING VIA RECTIFYING BATCH NORMALIZATION

## ABSTRACT

Quantization-aware Training (QAT) is able to reduce the training cost by quantizing neural network weights and activations in the forward pass and improve the speed at the inference stage. QAT can be extended to Fully-Quantized Training (FQT), which further accelerates the training by quantizing gradients in the backward pass as backpropagation typically occupies half of the training time. Unfortunately, gradient quantization is challenging as Stochastic Gradient Descent (SGD) based training is sensitive to the precision of the gradient signal. Particularly, the noise introduced by gradient quantization accumulates during backward pass, which causes the exploding gradient problem and results in unstable training and significant accuracy drop. Though Batch Normalization (BatchNorm) is a de-facto resort to stabilize training in regular full-precision scenario, we observe that it fails to prevent the gradient explosion when gradient quantizers are injected in the backward pass. Surprisingly, our theory shows that BatchNorm could amplify the noise accumulation, which in turn hastens the explosion of gradients. A BatchNorm rectification method is derived from our theory to suppress the amplification effect and bridge the performance gap between full-precision training and FQT. Adding this simple rectification loss to baselines generates better results than most prior FQT algorithms on various neural network architectures and datasets, regardless of the gradient bit-widths used (8,4, and 2 bits).

## 1 INTRODUCTION

Quantization-aware Training (QAT) is a popular track of research that simulates the neural network quantization (weights and activations) during the course of training to curb the inference-time accuracy drop of low-bit models (*e.g.* INT8 quantization). On the other hand, theoretical calculations on the BitOps (Yang & Jin (2021); Guo et al. (2020)) computation costs can easily conclude that backpropagation accounts for half of the computations during training. Empirical data[1] shows backward pass sometimes even costs more in practice. Decreasing the gradient bit-widths will apparently reduce computation overheads of backpropagation Horowitz (2014). If variables in backward pass are also quantized, adding up the forward quantization in QAT, all the network variables required in training would be fully quantized and the whole training process could be accelerated on dedicated hardware, *i.e.*, Fully-Quantized Training (FQT), providing huge accessibility of large model training to users with limited computation capability. Recent work Zhu et al. (2020) has shown that INT8 FQT speeds up the forward pass and the backward pass by $1.63\times$ and $1.94\times$ respectively when training ResNet-50 on ImageNet with NVIDIA Pascal GPU.

Yet gradient quantization under the FQT scheme is vastly underexplored, as it is notoriously more challenging than forward quantization in QAT. It is observed that network training is sensitive to the precision of gradients, and low-bit gradient quantization leads to unstable training and significant accuracy drop (see Fig. 1). More importantly, the accumulation of gradient quantization noise in backward pass (see Fig. 2) causes the exploding gradient problem during backpropagation, even resulting in training failure. In contrast to weight/activation quantization, gradient quantization noise produced during backpropagation cannot be automatically corrected by optimizing objective loss.

Unlike prior works on optimizing gradient quantizers for quantization noise reduction Zhou et al. (2016); Choi et al. (2018); Zhu et al. (2020), this paper reveals the negative effect of Batch Nor-

---

[1]https://github.com/jcjohnson/cnn-benchmarks

malization (BatchNorm) on amplifying the gradient quantization noise accumulation, when training deep Convolutional Neural Networks (CNNs) with low bit gradients. We show that the noise amplification effect further explodes the gradients during the backward pass. We thus propose a BatchNorm rectification method to suppress the noise amplification effect and alleviate the gradient explosion problem, which in turn enables stabilized training and better accuracy at low-bit gradients.

Our contributions are summarized as follows:

- We discover that BatchNorm fails to prevent the exploded low-bit gradients in full-quantized training through theoretic analysis, and may even amplify the accumulated gradient quantization noise, which further aggravates the gradient explosion.
- According to our theory, we propose a simple yet effective BatchNorm variance rectification algorithm without introducing noticeable overhead, to suppress the noise amplification effect, resulting in alleviated gradient explosion.
- Extensive experiments on MNIST, CIFAR-10, and ImageNet show that our method achieves improved training and higher accuracy over state-of-the-arts with vanilla gradient quantizers, regardless of the gradient bit-widths used (8,4,2 bits).

## 2 RELATED WORK

**Quantization-Aware Training (QAT)**. DoReFa-Net Zhou et al. (2016) proposed to optimize the clipping value and the scaling factor of the uniform quantizers for weights and activations separately. It was validated on image classification task under multiple bit-widths, but only with the rather simple AlexNet architecture. PACT Choi et al. (2018) proposed to quantize the activations with a learnable layer-wise clipping value, which not surprisingly achieved better accuracy than DoReFa-Net at 5-bit down to 2-bit. Most QAT works quantize weights and activations simultaneously, by optimizing the uniform quantization parameters Zhang et al. (2018a); Esser et al. (2020); Bhalgat et al. (2020), layer-wise or channel-wise mixed-precision quantization Jin et al. (2020); Lou et al. (2020), or leveraging non-uniform quantization such as Logarithmic quantizer Miyashita et al. (2016) and piece-wise linear quantizer Fang et al. (2020). Most recent QAT works Zhou et al. (2016); Choi et al. (2018); Zhang et al. (2018a); Esser et al. (2020); Bhalgat et al. (2020) used "Straight-Through Estimator" (STE) Bengio et al. (2013) to estimate the gradient of the non-differentiable quantization function, while other work Gong et al. (2019) softened the linear quantization operation in order to match the true gradient with STE.

**Fully-quantized Training (FQT)**. FQT aims to accelerate and quantize the backward pass of network training with low-bit error signals and gradients, agnostic to single machine or parallel training. Early attempt Zhou et al. (2016) adopted a primitive quantizer design based on uniform quantizer for gradients (without scaling and other optimization) and large performance drops are witnessed when training with low-bit gradients. SBM Banner et al. (2018) adopted fixed-point 8-bit gradient quantization, but only focused on improving the quantization schemes in the forward pass. WAGEUBN Yang et al. (2020) quantized gradients to 8-bit integers, but also showed a huge performance gap against its full-precision counterpart. NITI Wang et al. (2020) integrated gradient calculations with parameter update operations to reduce the gradient quantization noise with well-designed quantizers. However, it can only support shallow CNN architectures and did not explore any deeper networks with BatchNorm. In Zhu et al. (2020), the authors considered the sharp and wide distribution of gradients, and proposed to clip the gradients according to the deviation of the gradient distribution before quantization, achieving on-par results with full-precision training. To compensate the quantization loss on gradient, AFP Zhang et al. (2020) and CPT Fu et al. (2021) used higher precision data to aid low-precision training. DAINT8 Zhao et al. (2021) adopted a bespoke 8-bit channel-wise gradient quantization to suppress the negative effect of quantization noise during training. Gradient quantization with less than INT8 representations remains largely unexplored. FP4 Sun et al. (2020) managed to train modern CNN architectures using 4-bit gradients without significant accuracy loss, but the gradients were represented as floating-point numbers.

**Batch Normalization in QAT**. Most QAT approaches either left BatchNorm in between parameterized layer (Conv/FC) and activation layer (ReLU) without quantization Zhou et al. (2016); Choi et al. (2018) or with quantization Yang et al. (2020), or absorbed BatchNorm into Conv before weight quantization in the forward pass Jacob et al. (2018), or directly trained a BatchNorm-free shallow network architecture to achieve full 8-bit integer-only arithmetic Wang et al. (2020). To our

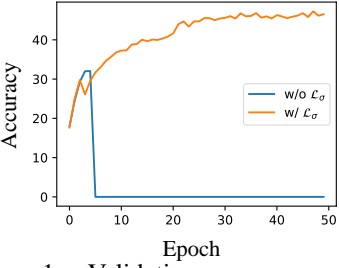

Figure 1: Validation accuracy curve of AlexNet on ImageNet with W2A2d$x$3d$W$2 w/ or w/o the proposed $\mathcal{L}_\sigma$. (d$x$ denotes error signal, d$W$ is gradients of weights, same below).

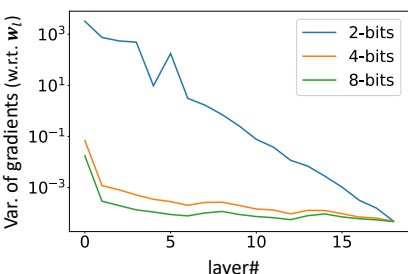

Figure 2: Illustration of gradient explosion problem (measured by the variance of gradients across layers) with $\{2, 4, 8\}$-bit uniformly quantized gradients. Statistics are computed from ResNet-20 on CIFAR-10 with 8-bit W/A. Y-axis is in log scale.

best knowledge, this paper is the first to systematically study the effect of BatchNorm on the training stability of quantized networks with low-bit gradients.

**Norm-free Networks**. There is another interesting line of works that gets rid of normalization layers from CNN architectures for good Zhang et al. (2018b) while still manages to train the full-precision networks stably. However, there is still a lack of attempts to adapt norm-free networks to QAT or even FQT settings at the first place, where potential problems need to be addressed when low-bit quantizers are introduced. Therefore, we decide currently it is not mature enough to discuss this track in this paper which targets at FQT.

**Prior efforts reducing variances in backpropagation**. Rectifying gradient variances during backpropagation has been sporadically discussed for full-precision training, *e.g.* in Kaiming Initialization He et al. (2015) where it leverages weight distributions in Conv and FC layers and one can opt for backward variance rectification if the gradients are observed to be chaotic. However, optimal forward and backward rectification still cannot be satisfied simultaneously, especially when backward signals contains significant amount of quantization noises. In our attempts, using "fan-out" mode in kaiming initialization alone still cannot avoid training crashes in worse cases (*e.g.* MobileNet-V2 under W4A4G4). To our knowledge, there is no work studying the impact of normalization layers in CNNs for gradient rectification under the FQT setting.

## 3 PRELIMINARIES

**Key Notations**. We denote the variance of a probabilistic variable as $D(\cdot)$. We denote gradient w.r.t. weights as $\boldsymbol{g_w}$ and error signal as $\boldsymbol{g_x}$. We use the plural term "gradients" to generally refer to all backward pass variables including $\boldsymbol{g_w}$ and $\boldsymbol{g_x}$. In such contexts, subscripts of $\boldsymbol{g}$ are omitted. Similar to the additive quantization noise for weight and activation quantization in Meller et al. (2019), the quantized gradients $\tilde{\boldsymbol{g}}$ at each layer can be decomposed into three parts:

$$\tilde{\boldsymbol{g}} = \boldsymbol{g} + e(\boldsymbol{g}) + \delta_q(\boldsymbol{g}), \tag{1}$$

where $\boldsymbol{g}$ and $e(\boldsymbol{g})$ denote the original gradients and the gradient quantization noise at current layer respectively, and $\delta_q(\boldsymbol{g})$ represents the accumulated gradient quantization noise propagated from **all** its succeeding layers during backward. E.g., for the $(l-1)^{th}$ layer, $\delta_q(\boldsymbol{g_{a_{l-1}}})$ represents the quantization noise accumulated from layer $l$ to the last layer (see Fig. 3).

In this manuscript, we format the bit-widths of weights (W), activations (A), backward errors (d$x$) and gradients of weights (d$W$) used in experiments as W/A/d$x$/d$W$ if error d$x$ and gradients d$W$ are assigned different bit-widths, or simplified as W/A/G if d$x$ and d$W$ have the same bit-width.

**Batch Normalization**. Batch Normalization Ioffe & Szegedy (2015) is widely adopted technique to stabilize the training of deep full-precision networks. The forward pass in a BatchNorm layer consists of operations calculating the mean and variance of each channel over a mini-batch with N input samples $\{\boldsymbol{x}_n\}_{n=1}^N$. Each channel of the input $\boldsymbol{x}_n$ is first normalized to $\hat{\boldsymbol{x}}_c = (\boldsymbol{x}_c - \mu_c)/\sigma_c$. The normalized input $\hat{\boldsymbol{x}}$ is finally linearly scaled and shifted as $\boldsymbol{y} = \boldsymbol{\gamma}^\intercal \hat{\boldsymbol{x}} + \boldsymbol{\beta}$. The relationship between backward error on BatchNorm's input $\boldsymbol{x}_n$ and that on $\hat{\boldsymbol{x}}_n$ is: (we omit channel notations here for simplicity)

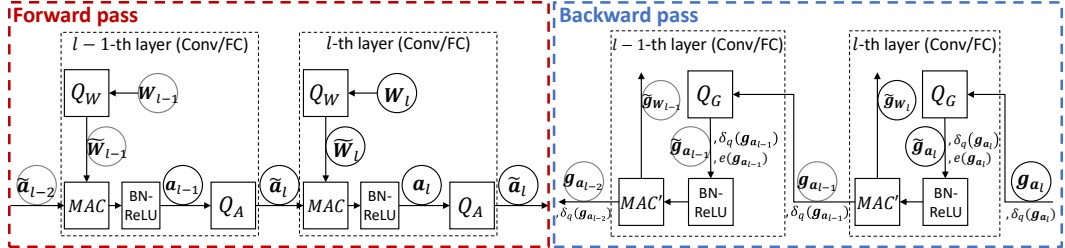

Figure 3: Illustration of weight/activation quantization in forward pass and gradient quantization in backward pass. MAC denotes the multiply–accumulate operations. $Q_{(.)}$ denotes the quantizer for weights (W), activations (A), or Gradients (G). In backward pass, the gradient quantization noise $\delta_q(\boldsymbol{g}_{\boldsymbol{a}_{l-1}})$ accumulated from the $l^{th}$ layer to the last layer is propagated to the $(l-1)^{th}$ layer and added to the $(l-1)^{th}$ layer's gradient quantization noise $e(\boldsymbol{g}_{\boldsymbol{a}_{l-1}})$ induced by itself.

$$\boldsymbol{g}_{\boldsymbol{x}_n} = \frac{1}{N\sigma}\left[ N\boldsymbol{g}_{\hat{\boldsymbol{x}}_n} - \sum_{n=0}^{N}\boldsymbol{g}_{\hat{\boldsymbol{x}}_n} - \hat{\boldsymbol{x}}_n\sum_{n=0}^{N}\boldsymbol{g}_{\hat{\boldsymbol{x}}_n}\hat{\boldsymbol{x}}_n \right].$$ (2)

## 4 PROBLEM IDENTIFICATION: ACCUMULATION OF GRADIENT QUANTIZATION NOISE EXPLODES GRADIENTS

Gradients play a crucial role in backpropagation based optimization and make a huge impact on training stability and convergence speed. Intuitively, as we inject quantizers in between backward pass, the error signal become more and more noisy each time it passes through a quantizer. As the bit-width decreases, the quantization noise injected in error signal at each single layer increases exponentially. From a perspective of variance, since quantization introduces additive noise $e(\boldsymbol{g})$ to the original signal $\boldsymbol{g}$, the variance of quantization noise $D(e(\boldsymbol{g}))$ is also added to the error signal $D(\boldsymbol{g})$, which will be reflected in the propagated errors and will increase over the course of backpropagation. As shown in Fig. 2, under various bit-width of gradients, we observe the variance of quantized error signals expanded during backpropagation. As the bit-width decreases, the variance also inflated much more drastically on shallow layers than late layers, implying that quantization noise is the culprit of the variance accumulation and explosion. As a result, the quantization impact on error signal and the weight update are severely affected, especially in those early layers. Eventually in worse cases when the accumulated quantization noise is overwhelming, the training goes off nature course and crashes. (*e.g.* under extremely low bit-width Fig. 7)

Fig. 3 provides a glance at the accumulation mechanism of the gradient quantization noise in the backward pass. During backpropagation, the quantization of the error signal on the $l^{th}$ layer introduces the quantization noise denoted as $e(\boldsymbol{g}_{\boldsymbol{a}_l})$. $e(\boldsymbol{g}_{\boldsymbol{a}_l})$ is propagated to its predecessor - the $(l-1)^{th}$ layer, together with the gradient quantization noise $\delta_q(\boldsymbol{g}_{\boldsymbol{a}_l})$ accumulated from the $(l+1)^{th}$ layer to the last layer $L$. Similarly, $e(\boldsymbol{g}_{\boldsymbol{a}_{l-1}})$ and $\delta_q(\boldsymbol{g}_{\boldsymbol{a}_{l-1}})$ are propagated to the $(l-2)^{th}$ layer, and so on.

As shown in Fig. 3, both forward and backward pass have similar accumulation phenomenon, but why backward pass suffers more from the quantization during training? This is because quantization noise introduced in the forward pass are reflected in the computation graph w.r.t. the objective loss, therefore their impact can be partially offset by quantization-aware training, while quantization noise introduced during backward does not contribute to the objective loss.

In view of some cases under the setting of distributed gradient compression Alistarh et al. (2017), which only quantizes full-precision gradients **after** backpropagation is done, can stably train CNNs with as low as 4-bit gradients, we attribute the exploding gradient problem in FQT setting mainly to the accumulation of gradient quantization noise introduced by low-bit gradient quantizers.

## 5 BATCHNORM AMPLIFIES THE ACCUMULATED NOISE

In this section, we develop a theoretical framework to understand the role of BatchNorm in gradient quantization, explaining why BatchNorm may worsen the gradient explosion problem in FQT. One can refer to Appendix A.2 for the proofs of the theorems.

**Quantifying the impact of BatchNorm on Noise Accumulation**. Through theoretical studies, we find that BatchNorm may amplify the accumulation of gradient quantization noise. This finding might be counter-intuitive as BatchNorm has been expected to regularize the "variance" and prevent the gradient explosion problem, by scaling the activations in forward pass. However, not only vanilla BatchNorm mainly focuses on rectifying variances in forward pass, but also does not count in the situation where the gradient signals are noisy. When error signal passes through such scaling layer inside BatchNorm, the error signal is also scaled by the *reciprocal* of the corresponding scaling factor ($\sigma$) when calculating its derivative. When training process is in full-precision, such scaling on error signal is manageable. But when the error signal contains accumulated noise from previous layers, the noise is scaled at the same time, causing unpredictable behavior to the backpropagation. Hence our theoretical focus is fundamentally different from previous "variance rectification and reduction" studies.

The following theorem quantifies how specifically BatchNorm affects the accumulation effect.

**Assumption 1.** *$\delta_q(\boldsymbol{g}_{\hat{\boldsymbol{x}}_i})$ and $\hat{\boldsymbol{x}}_i$ are i.d.d. and are both zero-mean Zhao et al. (2021).*

**Theorem 1.** *Given Assumption 1, for a BatchNorm layer in a quantized network, the relationship between the gradient quantization noise w.r.t. the BatchNorm's input $\boldsymbol{x}_i$ and that of the normalized input $\hat{\boldsymbol{x}}_i$ depends on the $\sigma$ of BatchNorm with batch size $N$, in the form of*

$$\eta = \frac{D(\delta_q(\boldsymbol{g}_{\boldsymbol{x}_i}))}{D(\delta_q(\boldsymbol{g}_{\hat{\boldsymbol{x}}_i}))} = \frac{1}{N^2\sigma^2}(N^2 + 2N). \tag{3}$$

**Remark 1.1.** *In Eq. (3), we define the amplification factor of the accumulated noise as $\eta$, which is the ratio of statistical variances between the scaled error signal $\delta_q(\boldsymbol{g}_{\boldsymbol{x}_i})$ and one before scaling $\delta_q(\boldsymbol{g}_{\hat{\boldsymbol{x}}_i})$ (see Fig. 4).*

**Corollary 1.1.** *To prevent BatchNorm from introducing more gradient quantization noise (i.e. the statistical variance $D(\delta_q(\boldsymbol{g}_{\hat{\boldsymbol{x}}_i}))$) when propagating the error signal to preceding layers, a desirable $\eta^*$ should not be greater than 1. Thus, a desirable $\boldsymbol{\sigma}$ of the BatchNorm should be $\boldsymbol{\sigma} \geq \sigma^* = \sqrt{1 + \frac{2}{N}}$.*

In short, Theorem 1 studies the amplification factor $\eta$ over the accumulated gradient quantization noise when gradient passes through BatchNorm scaling, and Corollary 1.1 provides a close form solution of the ideal condition of the BatchNorm variance $\sigma$ to satisify minimum noise accumulation.

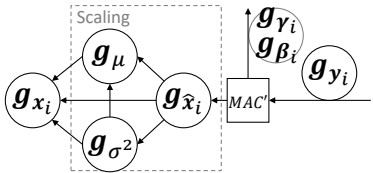

As the accumulated quantization noise during backward cannot be automatically amortized by the training objective in FQT, ones are left with only options to either (1) minimize the primary source of noise by improving gradient quantizer design, or (2) minimize the accumulation of such noise. With the first choice being obvious, in this paper, we instead aim to

Figure 4: Illustration of gradient propagation inside BatchNorm.

raise people's awareness about the second choice and the importance of properly scaling the noisy error signal to alleviate noise amplification problem in backward pass and the eventual training stability.

## 6 OUR METHOD: RECTIFYING BATCHNORM FOR GRADIENT QUANTIZATION

Inspired by our theory in Sec. 5, we develop a solution to suppress the noise amplification effect in a principled way, which in turn reduces the gradient explosion for improved training of quantized networks with low bit gradients. Based on Theorem 1 and Corollary 1.1, we expect the $\boldsymbol{\sigma}$ to be larger than the ideal value $\sigma^*$, so that the noise amplification factor $\eta = \frac{D(\delta_q(\boldsymbol{g}_{\boldsymbol{x}_i}))}{D(\delta_q(\boldsymbol{g}_{\hat{\boldsymbol{x}}_i}))}$ between the output and the input of BatchNorm is minimized. Therefore, we propose a method to stabilize the training with low bit gradients, by rectifying the variance of BatchNorm computed in forward pass:

$$\textbf{min.} \quad f(\boldsymbol{w})$$

$$\textbf{s.t.} \quad \mathcal{L}_{\boldsymbol{\sigma}} = \frac{1}{L}\sum_{l=1}^{L} \text{MSE}\left(\min\left(\frac{\boldsymbol{\sigma}_l}{\sigma^*}, 1\right), 1\right) = 0, \tag{4}$$

where $f : \mathbb{S} \to R$ is the regular objective loss (*e.g.*, cross-entropy loss for classification) and $\boldsymbol{w} \in \mathbb{S}$ denotes neural network weights. $\boldsymbol{\sigma}_L = \{\boldsymbol{\sigma}_l\}_{l=1}^{L}$ is the set of $\boldsymbol{\sigma}$ from all BatchNorm layers in the network with depth $L$. We use Mean Square Error (MSE) between $\frac{\sigma_l}{\sigma^*}$ and 1 to enforce that $\boldsymbol{\sigma}_l$ at each BatchNorm layer approaches to $\sigma^*$. The Lagrangian dual approximation form of Eq. (4) is:

$$f'(\boldsymbol{w}) = f(\boldsymbol{w}) + \lambda \mathcal{L}_{\boldsymbol{\sigma}}, \tag{5}$$

where $\lambda$ is an adjustable parameter to balance $f(\boldsymbol{w})$ and the proposed rectification loss $\mathcal{L}_{\boldsymbol{\sigma}}$.

**Gradient Computation and Computation Overhead**. The overhead introduced by our proposed rectification term only has **linear time complexity**. During backpropagation, the error signal of rectification term $\mathcal{L}_{\sigma}$ when propagated to BatchNorm's input $\boldsymbol{x}$ on channel $i$ at layer $l$ is:

$$\frac{\partial \mathcal{L}_{\sigma}}{\partial \boldsymbol{a}_l^i} = \begin{cases} \frac{2}{\sigma^* N}(\frac{1}{\sigma^*} - \frac{1}{\sigma_l^i})(\boldsymbol{a}_l^i - \mu_l^i), & \sigma_l^i < \sigma^*; \\ 0, & otherwise \end{cases}, \tag{6}$$

where $\sigma^*$, $N$ and channel-wise BatchNorm parameters $\mu_l^i, \sigma_l^i$ are all constant scalars during backpropagation, showing the added rectification is very cheap and negligible when training on devices capable of vectorized computation optimization including GPUs. More detailed analysis can be found in Appendix A.4.

# 7 EXPERIMENTS

To evaluate our method, we conduct extensive experiments on various neural network architectures and popular datasets for image classification with low-bit gradients.

**Experimental Setup**. To highlight the impact of BatchNorm, we only use two vanilla quantizers for gradients in all experiments without any optimizations: uniform quantizer and logarithmic quantizer (see Appendix A.1). Our method introduces only one hyper-parameter $\lambda$ in Eq. (5), which is manually initialized then can be ramped down by the cosine rule during training or stay the same. For each parametrized layer in backpropagation, the gradient $\boldsymbol{g_w}$ and error signal $\boldsymbol{g_x}$ are quantized separately, and they can be quantized to different bit-widths. To evaluate our rectification method, we ensure all the used CNN architectures have BatchNorm layers, including ShallowNet and AlexNet-BN, the architecture details of which are listed in Appendix A.3. More details are listed in Appendix A.4

## 7.1 MAIN RESULTS

**INT8 comparisons**. We compare our method (training with $\mathcal{L}_{\boldsymbol{\sigma}}$) to state-of-the-art gradient quantization approaches reporting results with 8-bit gradients: UI8 Zhu et al. (2020), FP8 Wang et al. (2018), AFP Zhang et al. (2020), SBM Banner et al. (2018), DAINT8 Zhao et al. (2021). Using 8-bit Logarithmic quantizer for gradients and the proposed rectifier $\mathcal{L}_{\boldsymbol{\sigma}}$, Tab. 1 shows that our method outperforms the state-of-the-arts in almost all cases, despite that we simple use vanilla quantizers on the gradient, while the quantizer designs of the counterparts are heavily engineered, *e.g.* DAINT8 Zhao et al. (2021) adopts vector quantization to process error signal in channel-wise manner. We found that MobileNet-V2 on ImageNet is harder to train with 8-bit gradients with vanilla quantizer designs even training with our $\mathcal{L}_{\boldsymbol{\sigma}}$, ending up with around $1\%$ accuracy drop than SOTA Zhao et al. (2021). As an ablation, for ResNet-20 on CIFAR10, our method boosts the accuracy by $1\%$ from baseline training (w/o $\mathcal{L}_{\boldsymbol{\sigma}}$), also for ResNet-18 on ImageNet, $\mathcal{L}_{\boldsymbol{\sigma}}$ achieves almost $2\%$ improvement.

**Comparisons on 4-bits gradients**. Since there are very few works on quantized neural networks with less than 8-bit gradients, we compare our method to a 4-bit floating-point quantization method named FP4 Sun et al. (2020) which actually adopt 4 bit floating-point representations, and the baseline which is without the proposed $\mathcal{L}_{\boldsymbol{\sigma}}$. As shown in Tab. 2, we observe that in most cases our method with INT4 gradient quantization reports higher accuracy than FP4 Sun et al. (2020), despite FP4 Sun et al. (2020) adopts floating-point quantization with customized radix and scaling selections. We also observe that our method (w/ $\mathcal{L}_{\boldsymbol{\sigma}}$) performs better than the baseline, further verifies the effectiveness of the proposed $\mathcal{L}_{\sigma}$. On the other hand, we observe MobileNet-V2 with INT4 gradients is still unstable and hard to converge even with our rectification method deployed (it explained why most FQT works did not report results for INT4 gradients on MobileNet-V2). Thus, we'd like to leave the FQT of MobileNet-V2 for future work.

| Dataset | Arch | Method | Top-1 |
|---|---|---|---|
| CIFAR-10 | ResNet-20 | UI8 | 92.0 |
| | | FP8 | 92.2 |
| | | DAINT8 | 92.8 |
| | | w/o $\mathcal{L}_\sigma$ | 91.9 |
| | | Ours | **92.9** |
| | Mobile-V2 | UI8 | 93.4 |
| | | DAINT8 | 94.4 |
| | | w/o $\mathcal{L}_\sigma$ | 94.5 |
| | | Ours | **94.6** |
| ImageNet | ResNet-18 | UI8 | 69.7 |
| | | FP8 | 67.3 |
| | | DAINT8 | 70.2 |
| | | SBM | 69.6 |
| | | w/o $\mathcal{L}_\sigma$ | 69.1 |
| | | Ours | **70.9** |
| | ResNet-50 | UI8 | 76.3 |
| | | SBM | 76.3 |
| | | DAINT8 | 76.6 |
| | | AFP | 76.2 |
| | | w/o $\mathcal{L}_\sigma$ | 76.4 |
| | | Ours | **76.8** |
| | Mobile-V2 | UI8 | 71.2 |
| | | DAINT8 | **71.9** |
| | | AFP | 70.5 |
| | | w/o $\mathcal{L}_\sigma$ | 70.74 |
| | | Ours | 70.9 |

Table 1: Comparing state-of-the-art methods with bit-width W8A8G8. (Mobile-V2=MobileNet-V2)

| Dataset | Arch | Method | Top-1 |
|---|---|---|---|
| CIFAR-10 | VGG-16 | FP4 | 91.5 |
| | | w/o $\mathcal{L}_\sigma$ | 90.7 |
| | | Ours | **92.5** |
| | ResNet-18 | FP4 | 92.7 |
| | | w/o $\mathcal{L}_\sigma$ | 93.7 |
| | | Ours | **94.0** |
| ImageNet | AlexNet-BN | FP4 | 56.3 |
| | | w/o $\mathcal{L}_\sigma$ | 57.2 |
| | | Ours | **57.3** |
| | ResNet-18 | FP4 | 68.3 |
| | | w/o $\mathcal{L}_\sigma$ | 69.61 |
| | | Ours | **69.71** |
| | ResNet-50 | FP4 | 74.01 |
| | | w/o $\mathcal{L}_\sigma$ | 73.67 |
| | | Ours | **73.4** |

Table 2: Comparison with FP4 gradient quantiation with bit-width W4A4G4. Log-INT4 quantizer is used for gradients.

| $\lambda$ | 0.1 | 0.25 | 0.5 | 1 |
|---|---|---|---|---|
| Top-1 (%) | 68.6 | 68.4 | **70.1** | 67.0 |

Table 3: Effect of $\lambda$ for training quantized ResNet-18 on CIFAR-10 with bit-width W2A2G2.

## 7.2 EXTREMELY LOW BIT-WIDTHS

Despite scarce exploitation in the wild and the challenges, we further attempt to quantize gradients to even lower bit-widths, with different backbone network architectures, bit-widths combinations, and gradient quantizers, to further evaluate the theoretical capability of the proposed rectifier $\mathcal{L}_\sigma$. When gradients are quantized to very low bit-widths, we expect the training becoming extremely unstable as the quantization noise and eventually the accumulation effect becoming much severer. Considering the increased training instability, we perform *three independent trials* for each experiments and report the comprehensive scores as (mean±std).

**Simple network architectures**. As shown in Tab. 4, we first study the training of a two-layer quantized neural network ShallowNet on MNIST. We set the bit-width for weights, activations, and gradients as 2-bit (W2A2G2) and use Logarithmic quantizer for gradients. We observed that baseline training without $\mathcal{L}_\sigma$ crashed twice out of 3 repetitive runs, while the training with the proposed $\mathcal{L}_\sigma$ is stable throughout. In other words, our method outperforms the baseline by a large margin, improving the average accuracy by +54.4% (from 38.8% to 93.2%).

On larger dataset ImageNet, we also have the similar observation for AlexNet-Bn, where baseline method fails to train completely while our method with $\mathcal{L}_\sigma$ can train stably throughout three trials.

**More complex network architectures**. We further test the effectiveness of our method in FQT training of networks with more complex structures. We push the boundary of the lowest bit-widths settings we can achieve, as lowest as *e.g.* W2A2G2 on ResNet-18 in Tab. 5. We observed that the FQT on VGG-16 is slightly more sensitive to the quantization than ResNet-18, thus we have to set higher bit-widths for VGG-16. We also notice that on VGG-16, error $(\mathrm{d}x)$ requires more bit-width than gradients w.r.t. weights $(\mathrm{d}W)$. In all, our method performs consistently better than the baseline (w/o $\mathcal{L}_\sigma$) on all settings, in particular, the performance gain on VGG-16 is significant.

As an additional remark, we notice that compared to other models, ResNets are more robust against more quantization noise throughout our experiments as suggested in Tab. 2 and Tab. 5. We conjec-

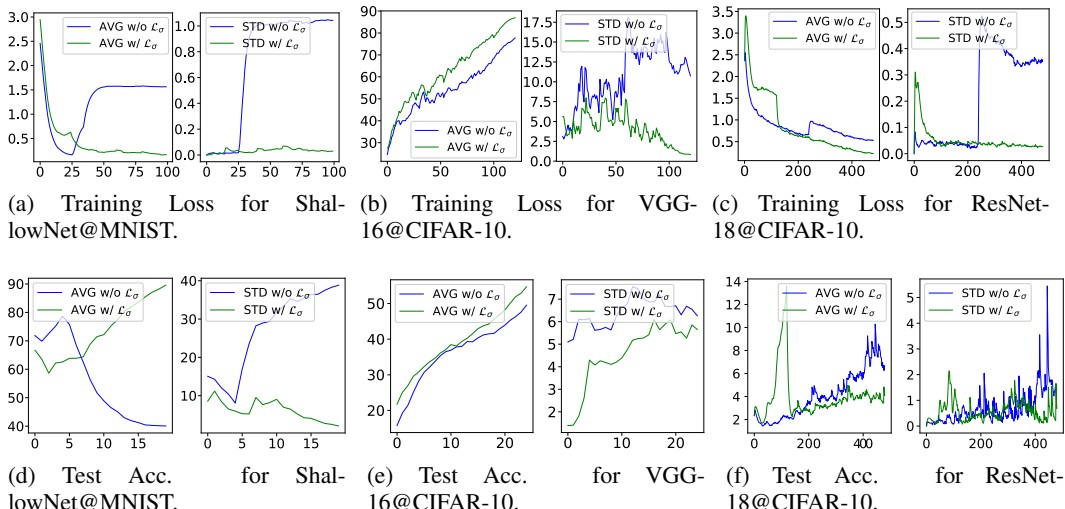

Figure 5: Illustration of stabilized training of quantized networks with lower than 4-bit gradients w/ $\mathcal{L}_\sigma$, compared to the baseline w/o $\mathcal{L}_\sigma$. Our method (w/ $\mathcal{L}_\sigma$) shows higher averaged accuracy and lower variance across repeated runs. X-axis in all sub-figures represents epochs.

| Bit-widths (W/A/d$x$/dW) | $\mathcal{L}_\sigma$ | Runs | | | Avg Top-1 (%) | Diff. (%) |
|---|---|---|---|---|---|---|
| | | #1 | #2 | #3 | | |
| ShallowNet on MNIST | | | | | | |
| 2/2/2/2 | w/o | 10.0 | 94.9 | 11.4 | $38.8 \pm 48.7$ | - |
| | w/ | 92.0 | 94.4 | 93.3 | $\mathbf{93.2 \pm 1.1}$ | $\mathbf{+54.4}$ |
| AlexNet-BN on ImageNet | | | | | | |
| 2/2/3/2 | w/o | - | - | - | - | - |
| | w/ | 47.6 | 49.46 | 46.53 | $\mathbf{47.86 \pm 1.48}$ | $\mathbf{+47.86}$ |

Table 4: Ablations of $\mathcal{L}_\sigma$ under extremely low bit-widths. Scores in red denotes failed train.

ture that it is due to the full-precision shortcuts within, making them naturally more robust against the accumulation effect during backward pass. Theoretical investigation towards this phenomenon would be an interesting future research topic.

### 7.3 OTHER DISCUSSIONS

**Can $\mathcal{L}_\sigma$ stabilize training?** Fig. 5 illustrates the improved training of quantized networks with less than 4-bit gradients, thanks to the rectification loss $\mathcal{L}_\sigma$. Fig. 5a and Fig. 5d are ShallowNet trained on MNIST for bit-width configuration W2A2G2 with Logarithmic gradient quantizer. Fig. 5b and Fig. 5e are VGG-16 trained on CIFAR-10 for W4A4d$x$4dW2 with Logarithmic gradient quantizer. Fig. 5c and Fig. 5f are ResNet-18 trained on CIFAR-10 for W4A4G4 with uniform gradient quantizer. We plot out the average loss/accuracy (AVG) and standard deviation of loss/accuracy (STD) of 3 trails separately. From Fig. 5a, Fig. 5b and Fig. 5c, we observe that the rectification loss $\mathcal{L}_\sigma$ dominates the training loss for the first few epochs, forcing the optimization adapted to the low-bit gradients. Afterwards, the training process becomes much more stable with training loss decreased and test accuracy increased gradually (see Fig. 5d, Fig. 5e and Fig. 5f). On the contrary, training without $\mathcal{L}_\sigma$ is not able to suppress the negative effect of gradient quantization noise, resulting in training instability or even crash.

**Can $\mathcal{L}_\sigma$ suppress the noise amplification effect?** To verify such stabilizing effect indeed comes from the proposed rectification, we further studies its impacts on gradient distribution layer by layer. Fig. 6 illustrates the distribution of variances of gradients w.r.t. layer output activations when training quantized VGG-16 on CIFAR-10 with bit-width W4A4d$x$4dW2. By injecting the rectification loss $\mathcal{L}_\sigma$ defined on BatchNorm in training objective function, one can see the noise amplification effect is largely suppressed, and thus the gradients are less exploded (measured by the variances of gradients).

| bit-widths (W/A/$\mathrm{d}x$/$\mathrm{d}W$) | Gradient Quantizer | $\mathcal{L}_\sigma$ | | Avg Top-1 (%) |
|---|---|---|---|---|
| VGG-16 | | | | |
| 4/4/4/2 | Uniform | w/o | | $50.0 \pm 8.4$ |
| | | w/ | | $\mathbf{66.7 \pm 8.7}$ |
| 4/4/4/2 | Logarithm | w/o | | $66.8 \pm 4.9$ |
| | | w/ | | $\mathbf{71.0 \pm 5.4}$ |
| ResNet-18 | | | | |
| 2/2/2/2 | Uniform | w/o | | $70.0 \pm 1.5$ |
| | | w/ | | $\mathbf{71.0 \pm 0.8}$ |
| 2/2/2/2 | Logarithm | w/o | | $59.9 \pm 2.1$ |
| | | w/ | | $\mathbf{61.6 \pm 2.0}$ |

Table 5: Ablations of $\mathcal{L}_\sigma$ and quantizers under extremely low bit-widths.

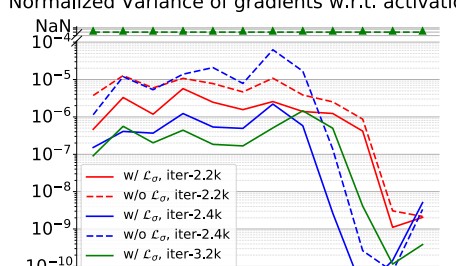

Figure 6: Distribution of variances of gradients w.r.t. activations when training VGG-16 on CIFAR-10 with bit-width W4A4d$x$4d$W$2. Logarithmic quantizer is used for gradients.

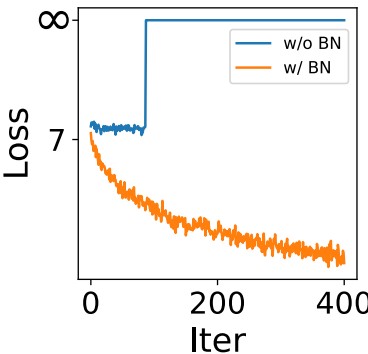

Figure 7: Train loss curve of AlexNet on ImageNet with W4A4d$x$4d$W$2 w/ or w/o Batch-Norm. (d$x$ denotes error signal, d$W$ is gradients of weights, same below).

As a result, it help prevent crashing in training with low bit gradients (see NaN at the 2.4k$^{th}$ iteration for training without $\mathcal{L}_\sigma$).

**Effect of** $\lambda$. As shown in Tab. 3, we study the effect of the hyper-parameter $\lambda$ when training quantized ResNet-18 on CIFAR-10 with bit-width W2A2G2. The optimal values of $\lambda$ for different experiments could be different and we heuristically tune them separately.

**Can BatchNorm layer be discarded when training with low-bit gradients?** Since BatchNorm amplifies the accumulated quantization noise during backpropagation, one may argue that a straightforward way to prevent the noise amplification effect is removing BatchNorm layer from the network architecture. To verify the point, we conducted experiments on training AlexNet with or without BatchNorm under W4A4d$x$4d$W$2 using logarithmic quantizer. As shown in Fig. 7, training AlexNet without BatchNorm quickly collapsed in the early stage in training. This implies that at the moment BatchNorm is still an essential building block to deep CNNs training for its rectifying benefits mainly in forward pass, while our rectification method complementarily stabilizes the backward pass, at least in the case when quantization is applied especially in FQT.

## 8 CONCLUSIONS

In this paper, we study an under-explored factor causing the gradient explosion problem when training deep CNNs with low-bit gradients, from a theoretical perspective. Our theory sheds light on the negative effect of BatchNorm in amplifying the accumulated gradient quantization noise during backpropagation, which leads to unstable training or even crash. The theory inspires a simple yet effective method to stabilize FQT with low-bit gradients, which consistently brings performance gain on a wide range of CNNs and datasets compared to state-of-the-art FQT algorithms.

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

# A APPENDIX

## A.1 GRADIENT QUANTIZER CHOICES

For gradient quantization, in each layer, the full-precision error $\mathrm{d}x$ and gradient $\mathrm{d}W$ are quantized, then de-quantized, and the quantized error propagated back to the next layer during backward pass

Zhou et al. (2016); Zhu et al. (2020); Yang et al. (2020); Wang et al. (2018). Uniform quantizer is the default and common choice gradient quantization, where given the gradients $\boldsymbol{g} \in \{\mathrm{d}x, \mathrm{d}W\}$, the asymmetric uniform quantizer with bit-width $B$ and quantization levels ranging from $[a, b]$ can be formulated as:

$$\boldsymbol{g}_q = \mathrm{Quant}_u(\boldsymbol{g}', a, b, B) = \mathrm{round}\left(\mathrm{clip}(\boldsymbol{g}', a, b) \cdot \frac{2^{B-1}-1}{b-a}\right), \tag{7}$$

where $\boldsymbol{g}' = \boldsymbol{g} + \delta_q(\boldsymbol{g})$ (see Sec. 3 for details) and $\mathrm{clip}(\boldsymbol{x}, a, b) = \min(\max(\boldsymbol{x}, a), b)$. The quantized gradients are de-quantized as $\tilde{\boldsymbol{g}} = \boldsymbol{g}_q \cdot \frac{b-a}{2^{B-1}-1}$. Similarly, the symmetric uniform quantizer can be defined as $\mathrm{Quant}_s = \mathrm{Quant}_u(\boldsymbol{g}, -c, c, B)$ where the $c$ is the clipping value.

We also explored logarithmic quantizer Miyashita et al. (2016) in this paper for gradient quantization given by:

$$\boldsymbol{g}_q = \mathrm{Quant}_{log}(\boldsymbol{g}', c, B) = \begin{cases} \mathrm{sign}(\boldsymbol{g}') \cdot 2^{\mathrm{Quant}_u(\log_2|\boldsymbol{g}'|,\log_2(c)-2^{B-1},\log_2(c),B)}, & \boldsymbol{g}' \neq 0; \\ 0, & \boldsymbol{g}' = 0 \end{cases} \tag{8}$$

## A.2 QUANTIFYING THE AMPLIFICATION EFFECT IN GQNA

**Theorem 1.** *For a BatchNorm layer within a quantized network, the relationship between gradient quantization error of BatchNorm's input $\boldsymbol{x}_i$ and that of the normalized input $\hat{\boldsymbol{x}}_i$ only depends on the batch size $N$ and the $\sigma$ of BatchNorm, in the form of*

$$\frac{D(\delta_q(\boldsymbol{g}_{\boldsymbol{x}_i}))}{D(\delta_q(\boldsymbol{g}_{\hat{\boldsymbol{x}}_i}))} = \frac{1}{N^2\sigma^2}(N^2 + 2N). \tag{9}$$

*Proof.* First we extend the Eq.2 to its quantized counterpart:

$$\tilde{\boldsymbol{g}}_{\boldsymbol{x}_i} = \frac{1}{N\sigma}\left[N\tilde{\boldsymbol{g}}_{\hat{\boldsymbol{x}}_i} - \sum_{i=0}^{N}\tilde{\boldsymbol{g}}_{\hat{\boldsymbol{x}}_i} - \hat{\boldsymbol{x}}_i\sum_{i=0}^{N}\tilde{\boldsymbol{g}}_{\hat{\boldsymbol{x}}_i}\hat{\boldsymbol{x}}_i\right] \tag{10}$$

and

$$e(\boldsymbol{g}_{\boldsymbol{x}_i}) = \frac{1}{N\sigma}\left[Ne(\boldsymbol{g}_{\hat{\boldsymbol{x}}_i}) - \sum_{i=0}^{N}e(\boldsymbol{g}_{\hat{\boldsymbol{x}}_i}) - \hat{\boldsymbol{x}}_i\sum_{i=0}^{N}e(\boldsymbol{g}_{\hat{\boldsymbol{x}}_i})\hat{\boldsymbol{x}}_i\right]. \tag{11}$$

Expand Eq. (10) using Eq. (1):

$$\boldsymbol{g}_{\boldsymbol{x}_i} + \delta_q(\boldsymbol{g}_{\boldsymbol{x}_i}) + e(\boldsymbol{g}_{\boldsymbol{x}_i}) = \frac{1}{N\sigma}\left[N(\boldsymbol{g}_{\hat{\boldsymbol{x}}_i} + \delta_q(\boldsymbol{g}_{\hat{\boldsymbol{x}}_i}) + e(\boldsymbol{g}_{\hat{\boldsymbol{x}}_i})) - \sum_{i=0}^{N}(\boldsymbol{g}_{\hat{\boldsymbol{x}}_i} + \delta_q(\boldsymbol{g}_{\hat{\boldsymbol{x}}_i}) + e(\boldsymbol{g}_{\hat{\boldsymbol{x}}_i})) \right.$$
$$\left. -\hat{\boldsymbol{x}}_i\sum_{i=0}^{N}(\boldsymbol{g}_{\hat{\boldsymbol{x}}_i} + \delta_q(\boldsymbol{g}_{\hat{\boldsymbol{x}}_i}) + e(\boldsymbol{g}_{\hat{\boldsymbol{x}}_i}))\hat{\boldsymbol{x}}_i\right]. \tag{12}$$

Eliminating terms in Eq. (12) using Eq. (1) and Eq. (11), we have:

$$\delta_q(\boldsymbol{g}_{\boldsymbol{x}_i}) = \frac{1}{N\sigma}\left[N\delta_q(\boldsymbol{g}_{\hat{\boldsymbol{x}}_i}) - \sum_{i=0}^{N}\delta_q(\boldsymbol{g}_{\hat{\boldsymbol{x}}_i}) - \hat{\boldsymbol{x}}_i\sum_{i=0}^{N}\delta_q(\boldsymbol{g}_{\hat{\boldsymbol{x}}_i})\hat{\boldsymbol{x}}_i\right]. \tag{13}$$

Calculate the variance $D(\cdot)$ of LHS and RHS of the above we have: (assume $\delta_q(\boldsymbol{g}_{\hat{\boldsymbol{x}}_i})$ and $\hat{\boldsymbol{x}}_i$ are i.d.d. and are both zero-mean)

$$D(\delta_q(\boldsymbol{g}_{\boldsymbol{x}_i})) = \frac{1}{N^2\sigma^2}D\left(N\delta_q(\boldsymbol{g}_{\hat{\boldsymbol{x}}_i}) - \sum_{i=0}^{N}\delta_q(\boldsymbol{g}_{\hat{\boldsymbol{x}}_i}) - \hat{\boldsymbol{x}}_i\sum_{i=0}^{N}\delta_q(\boldsymbol{g}_{\hat{\boldsymbol{x}}_i})\hat{\boldsymbol{x}}_i\right) \tag{14}$$

$$= \frac{1}{N^2\sigma^2}\left[N^2D(\delta_q(\boldsymbol{g}_{\hat{\boldsymbol{x}}_i})) + D\left(\sum_{i=0}^{N}\delta_q(\boldsymbol{g}_{\hat{\boldsymbol{x}}_i})\right) + D\left(\hat{\boldsymbol{x}}_i\sum_{i=0}^{N}\delta_q(\boldsymbol{g}_{\hat{\boldsymbol{x}}_i})\hat{\boldsymbol{x}}_i\right)\right] \tag{15}$$

$$= \frac{1}{N^2\sigma^2}\left[N^2D(\delta_q(\boldsymbol{g}_{\hat{\boldsymbol{x}}_i})) + \sum_{i=0}^{N}D(\delta_q(\boldsymbol{g}_{\hat{\boldsymbol{x}}_i})) + D\left(\hat{\boldsymbol{x}}_i\sum_{i=0}^{N}\delta_q(\boldsymbol{g}_{\hat{\boldsymbol{x}}_i})\hat{\boldsymbol{x}}_i\right)\right] \tag{16}$$

$$= \frac{1}{N^2\sigma^2} \left[ N^2 D(\delta_q(\boldsymbol{g}_{\hat{\boldsymbol{x}}_i})) + N D(\delta_q(\boldsymbol{g}_{\hat{\boldsymbol{x}}_i})) + D(\hat{\boldsymbol{x}}_i) \sum_{i=0}^{N} D(\delta_q(\boldsymbol{g}_{\hat{\boldsymbol{x}}_i})) D(\hat{\boldsymbol{x}}_i) \right] \quad (17)$$

$$= \frac{1}{N^2\sigma^2} \left[ N^2 D(\delta_q(\boldsymbol{g}_{\hat{\boldsymbol{x}}_i})) + N D(\delta_q(\boldsymbol{g}_{\hat{\boldsymbol{x}}_i})) + N D(\delta_q(\boldsymbol{g}_{\hat{\boldsymbol{x}}_i})) \right] \quad (18)$$

$$= \frac{1}{N^2\sigma^2} \left[ N^2 D(\delta_q(\boldsymbol{g}_{\hat{\boldsymbol{x}}_i})) + 2N D(\delta_q(\boldsymbol{g}_{\hat{\boldsymbol{x}}_i})) \right] \quad (19)$$

$$= \left[ \frac{1}{\sigma^2} + \frac{2}{N\sigma^2} \right] D(\delta_q(\boldsymbol{g}_{\hat{\boldsymbol{x}}_i})). \quad (20)$$

Therefore,

$$\frac{D(\delta_q(\boldsymbol{g}_{\boldsymbol{x}_i}))}{D(\delta_q(\boldsymbol{g}_{\hat{\boldsymbol{x}}_i}))} = \frac{1}{\sigma^2} + \frac{2}{N\sigma^2} = \frac{1}{N^2\sigma^2}(N^2 + 2N). \quad (21)$$

$$\square$$

### A.3 ARCHITECTURE DETAILS

**ShallowNet**.

```python
class ShallowNet(nn.Module):
    def __init__(self):
        super().__init__()
        self.conv1 = nn.Conv2d(1, 20, 5, 1)
        self.bn1 = nn.BatchNorm2d(20)
        self.relu1 = nn.ReLU(inplace=False)
        self.pool1 = nn.MaxPool2d(2, 2)
        self.conv2 = nn.Conv2d(20, 50, 5, 1)
        self.bn2 = nn.BatchNorm2d(50)
        self.relu2 = nn.ReLU(inplace=False)
        self.pool2 = nn.MaxPool2d(2, 2)
        self.avgpool = nn.AvgPool2d(4, stride=1)
        self.fc = nn.Linear(50, 10)

    def forward(self, x):
        x = self.pool1(self.relu1(self.bn1(self.conv1(x))))
        x = self.pool2(self.relu2(self.bn2(self.conv2(x))))
        x = self.avgpool(x)
        x = x.view(x.size(0), -1)
        x = self.fc(x)
        return x
```

**AlexNet-BN**.

```python
class AlexNetBN(nn.Module):
    def __init__(self, num_classes=1000):
        super(AlexNetBN, self).__init__()
        self.features = nn.Sequential(
            nn.Conv2d(3, 96, kernel_size=12, stride=4),
            nn.ReLU(inplace=True),

            nn.Conv2d(96, 256, kernel_size=5, padding=2, groups=2,
            ↪  bias=False),
            nn.BatchNorm2d(256, eps=1e-4, momentum=0.9),
            nn.MaxPool2d(kernel_size=3, stride=2, ceil_mode=True),
            nn.ReLU(inplace=True),

            nn.Conv2d(256, 384, kernel_size=3, padding=1, bias=False),
            nn.BatchNorm2d(384, eps=1e-4, momentum=0.9),
            nn.MaxPool2d(kernel_size=3, stride=2, padding=1),
            nn.ReLU(inplace=True),

            nn.Conv2d(384, 384, kernel_size=3, padding=1, groups=2,
            ↪  bias=False),
```

```
            nn.BatchNorm2d(384, eps=1e-4, momentum=0.9),
            nn.ReLU(inplace=True),

            nn.Conv2d(384, 256, kernel_size=3, padding=1, groups=2,
            ↪  bias=False),
            nn.BatchNorm2d(256, eps=1e-4, momentum=0.9),
            nn.MaxPool2d(kernel_size=3, stride=2),
            nn.ReLU(inplace=True),
        )

        self.classifier = nn.Sequential(
            nn.Linear(256 * 6 * 6, 4096, bias=False),
            nn.BatchNorm1d(4096, eps=1e-4, momentum=0.9),
            nn.ReLU(inplace=True),
            nn.Linear(4096, 4096, bias=False),
            nn.BatchNorm1d(4096, eps=1e-4, momentum=0.9),
            nn.ReLU(inplace=True),
            nn.Linear(4096, num_classes),
        )

    def forward(self, x):
        x = self.features(x)
        x = x.view(x.size(0), 256 * 6 * 6)
        x = self.classifier(x)
        return x
```

## A.4 TRAINING SETTINGS AND HYPER-PARAMETERS

Following Sun et al. (2020), we adopt Parameterized Clipping Activation (PACT) Choi et al. (2018) for activation quantization and Statistics Aware Weight Binning (SAWB) Choi et al. (2019) for weight quantization. We choose the SGD optimizer and train all quantized network models from scratch. The learning rate is adjusted with a cosine scheduler (Loshchilov & Hutter (2017)). For MNIST, we set the learning rate as 0.1, weight decay as 0.0001, and train for 20 epochs. For CIFAR-10, we set the learning rate as 0.1 for all architectures, weight decay 0.0001 for ResNet-18 and VGG-16, and weight decay 0.0004 for MobileNet-V2. For ImageNet, we set the learning rate as 0.0512 and weight decay as 0.0001 for ResNet-18, learning rate 0.01 and weight decay 0.0005 for AlexNet, and learning rate 0.1 and weight decay 0.00004 for MobileNet-V2. All models on CIFAR-10 and ImageNet are trained for 120 epochs, except for MobileNet-V2 that is trained for 150 epochs. For ImageNet, training images are randomly cropped to $224 \times 224$ and then randomly flipped horizontally. For experiments on MNIST and CIFAR-10, we repeat the training of each model for 3 runs by varying the random seed and report the average accuracy with standard deviation. Same as other INT8 works Zhu et al. (2020), we left the first and the last weighted layers as well as activations in full-precision for all INT8 experiments. For INT4 experiments, we followed the setting in Choi et al. (2019) that only left shortcut layers in ResNets in full-precision and quantized all other layers.

We specify the choices of $\lambda$ and its ramping down strategy as below.

**Comparing SOTA with 8-bit gradients**

| Dataset | Arch | $\lambda$ | Ramping down? |
|---------|------|-----------|---------------|
| CIFAR-10 | ResNet-20 | 0.5 | N |
|          | MobileNet-V2 | 0.5 | N |
| ImageNet | ResNet-18 | 0.5 | N |
|          | ResNet-50 | 0.5 | N |
|          | MobileNet-V2 | 0.5 | N |

Table 6: Comparing state-of-the-art with 8-bit gradients.

**Comparing SOTA with 4-bit gradients**

**Experiments with lower-than-4-bit gradients**

| Dataset | Arch | $\lambda$ | Ramping down? |
|---|---|---|---|
| CIFAR-10 | VGG-16 | 0.5 | Y |
| | ResNet-18 | 0.5 | Y |
| ImageNet | AlexNet | 0.5 | N |
| | ResNet-18 | 0.5 | Y |

Table 7: Comparing our 4-bit fixed-point gradient quantization to the 4-bit floating-point gradient quantiation FP4 Sun et al. (2020).

| bit-widths (W/A/$\mathrm{d}x$/$\mathrm{d}W$) | Gradient Quantizer | $\lambda$ | Ramp down? |
|---|---|---|---|
| ShallowNet on MNIST | | | |
| 2/2/2/2 | Uniform | 0.1 | N |
| VGG-16 on CIFAR-10 | | | |
| 4/4/4/2 | Uniform | 0.5 | Y |
| 4/4/4/2 | Logarithm | 0.5 | Y |
| ResNet-18 on CIFAR-10 | | | |
| 2/2/2/2 | Uniform | 0.5 | Y |
| 2/2/2/2 | Logarithm | 0.1 | Y |

Table 8: Less than 4-bit gradients.

## A.5 DETAILED ANALYSIS OF COMPUTATION OVERHEAD

To more directly derive in the linear complexity conclusion, one can simplify Eq. (6) into $\frac{\partial \mathcal{L}_\sigma}{\partial \boldsymbol{a}_l} = \boldsymbol{C}_1 \odot \boldsymbol{a}_l + \boldsymbol{C}_2$, where $\odot$ denotes the element-wise multiplication, $\boldsymbol{C}_1, \boldsymbol{C}_2$ are both constant tensors with the same shape as $\boldsymbol{a}_l$. The partial derivative $\frac{\partial \mathcal{L}_\sigma}{\partial \boldsymbol{a}_l}$ is further aggregated with the error signal from objective function $\frac{\partial f(\boldsymbol{w})}{\partial \boldsymbol{a}_l}$ to compute the gradient w.r.t. weights $\frac{\partial f'(\boldsymbol{w})}{\partial \boldsymbol{w}_l}$. Here $\mathcal{L}_\sigma$ introduces no extra computation cost.

Tab. 9 gives an empirical verification of the actual training overhead of our method compared to the baseline.

| Dataset | Arch | w/o $\mathcal{L}_\sigma$ | Ours | Diff. |
|---|---|---|---|---|
| ImageNet | ResNet-18 | 43.4 | 44.6 | 0.67% |
| | ResNet-50 | 143.32 | 152.58 | 6.46% |
| | MobileNet-V2 | 162.4 | 171.2 | 5.41% |

Table 9: Comparison of training time (hours).

## A.6 DETAILED ABLATIONS ON HYPER-PARAMETER (ON HIGHER BIT-WIDTHS)

To comprehensively evaluate the influence of the choices of hyper-parameter $\lambda$, we further tested the model performance on higher bit-width.

Tab. 10 shows the results under different $\lambda$ values of ResNet-18 on CIFAR-10. The quantizer choice for gradients is log quantizer.

| $\lambda$ | 0 | 0.1 | 0.25 | 0.5 | 1 |
|---|---|---|---|---|---|
| Top-1 (%) | 93.7 | 93.88 | 93.57 | 94.0 | 93.71 |

Table 10: Ablation study of $\lambda$ under bit-widths W4A4G4.

