# OpenReview forum: "Improved Fully Quantized Training via Rectifying Batch Normalization"
_ICLR.cc/2023/Conference — Submitted to ICLR 2023_

### Official Review · Reviewer_GegD · 2022-10-24

**Confidence:** 3
**Correctness:** 3
**Technical Novelty And Significance:** 3
**Empirical Novelty And Significance:** 3
**Recommendation:** 5

**Clarity, Quality, Novelty And Reproducibility:**

The motivation seems novel, the empirical results are comprehensive, and the paper is easy to follow.
Codes are not provided, so the reproducibility cannot be confirmed.

**Strength And Weaknesses:**

Strength:
1. The observation of the batchnorm's negative effect on gradient explosion seems novel.
2. The empirical results are comprehensive.

Weaknesses:
1. Does theorem 1 take weight and activation quantization into consideration? Why is the result not affected by the quantization?
2. Corollary 1.1: should be not greater -> should not be greater
3. Please provide a formal definition of D in Theorem 1.
4. Eq. (4): why should $\sigma_l$ approaches $\sigma_*$? Theorem 1 seems to suggest that $\eta$ should be as large as possible.
5. From Table 4, the performance is extremely sensitive to the choice of $\lambda$, considering the gap in Table 1 and 2 is not very obvious compared to $\lambda$. Therefore, repeated experiments should be conducted for fair comparison in Table 1 and 2.
6. How does it work for training full-precision networks with quantized gradients?

**Summary Of The Paper:**

The author studies the batchnorm's negative effect when training with quantized gradient. To address the problem, the author proposes to add an additional loss to control the variance of batchnorm. Empirical results show improvement when quantizing weight, activation and gradient.

**Summary Of The Review:**

The authors make a novel observation that batchnorm may contribute to gradient explosion in FQT. But I think more comprehensive theoretical and empirical results may be needed to make it more convincing. The method proposed seem to be too sensitive to the choice of the new hyper-parameter $\lambda$, which makes it hard to validate its better performance since the improvements in Table 1 and 2 become negligible compared with the variance in Table 3.

---

> ### Author Response · Authors · 2022-11-09
> **Response to R3**
>
> Dear R3,
>
> We appreciate your comments and suggestions to our manuscript! Here we address them as below:
>
> 1. Does theorem 1 take weight and activation quantization into consideration? Why is the result not affected by the quantization?
> > Theorem 1 doesn’t explicitly take W and A quantization into account, as our focus in this work is quantization during backward. Here we wanted to discuss the behavior of the quantization error propagation and accumulation during backward, where forward variables are regarded as constants. Implicitly, however, the influences of W and A are already reflected in training loss as well as in the error signal.
> 2. Corollary 1.1: should be not greater -> should not be greater
> > Thanks for pointing this out, we have corrected it in the revised draft.
> 3. Please provide a formal definition of D in Theorem 1.
> > We have defined $D(e(\mathbf{g}))$ as the variance of quantization noise before the theorem in the manuscript. Following your suggestion, we have added the definition to the “Key Notations” in Section 3.
> 4. Eq. (4): why should σl approaches σ∗? Theorem 1 seems to suggest that η should be as large as possible.
> > In Eq.4, we want $\min(\frac{\mathbf{\sigma}_l}{\sigma^\ast}, 1)$ to approach 1, so only the case when $\mathbf{\sigma}_l > \sigma^\ast$ is punished, which is consistent with Theorem 1 and Corollary 1.1.
> 5. From Table 4, the performance is extremely sensitive to the choice of λ, considering the gap in Table 1 and 2 is not very obvious compared to λ. Therefore, repeated experiments should be conducted for fair comparison in Table 1 and 2.
> > We assume you are referring to the ablation of $\lambda$ in Table 3. Table 3 is under extremely low bit-widths (W2A2G2) as mentioned in the caption, so we expected the influence of the hyper-parameters to be obvious. Under higher bit-width, e.g. W4A4G4 in Table 2, the choice of $\lambda$ doesn’t impact the performance that much.
> For ablation of $\lambda$ under W4A4G4 (ResNet-18 on CIFAR-10, log quantizer), the experimental results are as below:
> | $\lambda$ | 0 | 0.1 | 0.25 | 0.5 | 1 |
> |  ----  | ----  | ---- | ---- | ---- | ---- |
> | Test Accuracy | 93.7 | 93.88 | 93.57 | 94.0 | 93.71 |
> We have added them in the __Appendix A.6__.
> 6. How does it work for training full-precision networks with quantized gradients?
> > Thanks for the very interesting suggestion. However, we cannot figure such setting where W and A are full-precision and only gradients are quantized to have much practical significance, because with the same computation budget, quantizing W and A and leaving gradients in full-precision are bound to result in better performance, with the reason stated in the beginning of Section 4.
> We want to remark that theoretically, our method is likely still going to fit in such setting under simulated FQT training, as our theory doesn’t depend on the precision of forward variables.
> ---
> Codes are not provided, so the reproducibility cannot be confirmed.
> > Publishing codes will require permission from our institute. We will disclose the codes once it’s approved.
>
> ---
>
> Please let us know if there are any unclear points remaining.
>
> Sincerely,
>
> Authors.

---

### Official Review · Reviewer_FcHZ · 2022-10-25

**Confidence:** 4
**Correctness:** 2
**Technical Novelty And Significance:** 1
**Empirical Novelty And Significance:** 3
**Recommendation:** 3

**Clarity, Quality, Novelty And Reproducibility:**

The quality of writing is low, with many typos and hard-to-parse sentences. The general presentation of the paper needs significant improvement. Some of the more notable issues are:
 - "On the other hand, theoretical calculations on the BitOps computation costs..." - I am unfamiliar with BitOps; is it some benchmark? In any case, it should be clarified what these are or at least a reference would be needed.
 - "Empirical data1 shows backward sometimes even costs more in practice." - I presume the authors mean backdrop.
 - In Eq 1, the $\mathbf{g_{x_n}}$ terms are undefined. Are they $g_x = \nabla_x \hat{x}$? Also, the second and third terms involving the sums contain mistakes, as the sums use $i$ as an index, but this index is not used in the summands. What is the relevance of the earlier defined $y$ symbol?
 - In Eqs 1 and 2, what is the difference between $g$ and $g_{x}$? Is $g$ the gradient operator, and $g_{x}$ is the gradient evaluated at $x$?
 - In Eq 2, I would avoid using $\Delta$ as it is usually reserved for the Laplacian operator.
 - "Gradients play a crucial role in back-propagation based optimization and make a huge impact on training stability and convergence speed." - I don't disagree with this statement, but I wonder if the authors might actually be referring to the error in the gradients.
 - "As the bit-width decreases, the quantization noise injected in error signal at each single layer increases exponentially." - This is a very confusing sentence, as it is unclear what the authors mean. In the following sentence, they explain that the error accumulates multiplicatively, and in the sentence after that, they finally explain what they mean by the exponential increase.
 - Figure 2 is not great at demonstrating the claimed exponential increase. I suggest the authors redesign it to plot bit width against gradient error since that is what they are analyzing.
 - The colour scheme in Figure 3 is confusing, especially the colour gradients for the $\delta$ terms.
 - The "Prior efforts reducing variances in backprop" paragraph in Section 5 should be moved (perhaps even verbatim) to the related works section, as it doesn't pertain to the proclaimed content of the section at all.
 - The symbol $D$ is used in the text as the "statistical variance" operator, but the authors' whole mathematical setup is deterministic, so it is unclear what they mean.
 - The font sizes of the labels of Figures 3 - 6 should be increased as they are currently difficult to read.
 - X-axis labels are missing in Figure 3
 - Table 3 is not refereced in the main text.

**Strength And Weaknesses:**

## Strengths
The authors propose a solution to a relevant problem, stabilizing FQT, which should be of interest to many people in the ML community. The proposed solution is simple to implement and is intuitively clear: the regularization term they propose doesn't allow the normalizing standard deviation term in BN to become "too small" for an appropriate notion of "small", which prevents instability during training. Finally, they compare their proposed approach on a reasonable selection of benchmarks and verify their method's effectiveness from different angles.

## Weaknesses
 - Simply by virtue of the extremely sloppy mathematical setup, Thm 1 cannot possibly be correct. The theorem makes a statement about the variance of deterministic quantities. Halfway through the proof in the appendix, the authors suddenly assume that these quantities are random and state assumptions on them. I am not claiming that the statement of Thm 1 is "morally wrong" because perhaps it can be shown to be correct with a proper setup. However, as the authors present it, it is uninterpretable.
 - Given how well the authors' solution appears to perform on the benchmarks they consider, I think it would be very useful to show the performance gap between models trained with their method or that are trained without using quantization.

**Summary Of The Paper:**

The authors consider fully quantized training (FQT), a subfield of quantization-aware training (QAT), where even the gradients are quantized during backprop. A common issue in FQT is that training can be very unstable.

The authors study the role of batch normalization (BN) in causing training instability, and they identify BN as a significant culprit. Hence, the authors propose adding a regularization term in the loss function of models using BN, and they empirically demonstrate their approach's effectiveness.

**Summary Of The Review:**

The authors propose a solution to a relevant problem and demonstrate its effectiveness empirically. However, the paper suffers heavily in terms of clarity of writing and mathematical formalism. Most importantly, it contains mathematical statements that are, at best uninterpretable and flat-out wrong at worst. Hence, the paper is unpublishable in its current form.

---

> ### Author Response · Authors · 2022-11-09
> **Response to R2**
>
> Dear R2,
>
> We appreciate your very detailed review and precious suggestions on our manuscript.
>
> ---
> __Doubts on Thm 1__
>
> Thanks for your careful read on our theorem and proof.
> About the determinacy, the gradients are typically regarded as random variables when we study them over many minibatches during training, so as the accumulated error $\delta\_q(\cdot)$ of them in Thm 1, since they are functions of random variables.
> In the proof, the independent and zero-mean assumptions are adopted from previous publications UI8 and DAINT8, which are shown to be close enough to the empirical statistics.
> We hope mentioning these assumptions earlier in the main text could better remind the readers that the gradients here are random variables.
>
> ---
>
> __Should compare to w/o $\mathcal{L}\_\sigma$__
>
> In the manuscript, we already showed the performance gap between w/ and w/o the proposed method, which are referred to as “Ours” and “w/o $\mathcal{L}_\sigma$” respectively.
>
> ---
>
> We regret that the current draft didn’t meet your standard, therefore we made significant revisions following your comments under “Clarity, Quality, Novelty And Reproducibility”. Specifically:
> - I am unfamiliar with BitOps. it should be clarified or at least a reference.
> > For each operation (e.g. matmul), BitOPs = OPs x [Product of all operands’ bit-widths], so when operands are all 32-bit, BitOPs=FLOPs. We added a reference [FracBits](https://arxiv.org/abs/2007.02017) in the revised version.
> - In Eqs 1 and 2, is g the gradient operator, and gx is the gradient evaluated at x?
> > Yes, you are correct. We have added an explanation of terms related to __g__ in the “Key Notations” part in Section 3.
> - In Eq 1, are the gxn terms =∇xx^? The 2nd and 3rd terms contain mistakes. What is the relevance of the earlier defined y symbol?
> > Yes, your understanding of $g$ is correct. Thanks for pointing out the typo in Eq.1, and we corrected it in revised version. $y$ is the BN output, we included it to give a thorough introduction to the background of BN.
> - "Gradients play a crucial role in back-propagation based optimization..." - I don't disagree with this statement, but are you actually referring to error of gradients instead?
> > Essentially we wanted to express that the **quality** of gradients is important for training stability, therefore the error in gradients would also play a part in it.
> - "As the bit-width decreases, the quantization noise injected in error signal at each single layer increases exponentially." - This sentencing and the following 2 sentences are very confusing.
> > We wish to clarify that (1) the error __itself__, at each layer, increases exponentially with the decreasing bit-width; (2) When bit-width is fixed, the __accumulation__ of error increases multiplicatively during backpropagation. We have rewritten these sentences to reduce the confusion.
> - Authors should redesign Fig 2 to plot bit-width vs gradient error since that is what they are analyzing.
> > Thanks for your kind advice. We also want to make a slight clarification here that exponential increase of error w.r.t. bit-width is not our main focus since the challenges of gradient quantization are already discussed in various prior works, so we don't intend to expand too much demonstrating it. We originally designed Fig 2 to demonstrate both exponential property and the accumulation property across layers during backprop. Therefore, we adjusted the analysis text of Fig 2 to hopefully weaken the portion on the exponential claim.
> - The symbol D is used in the text as the "statistical variance" operator, but the authors' whole mathematical setup is deterministic, so it is unclear what they mean.
> > As we explained, the stochastic gradients here are regarded as random variables instead of deterministic quantities.
> - X-axis labels are missing in Fig 3.
> > We assume you mean Fig 5. We didn't include them in since the sub-figures are already densely placed. We have added explanatory notes in the Fig 5 caption.
>
> For the remaining points regarding the writing and presentations, we made changes to the draft accordingly, and please check our revision summary in a separate response, thanks.
>
> ---
>
> We sincerely hope this response and changes can resolve some of your concerns. We would be very happy to continue the discussion!
>
> Sincerely,
>
> Authors.

---

### Official Review · Reviewer_dU93 · 2022-10-25

**Confidence:** 2
**Correctness:** 4
**Technical Novelty And Significance:** 4
**Empirical Novelty And Significance:** 3
**Recommendation:** 6

**Clarity, Quality, Novelty And Reproducibility:**

Clarity: The paper is easy to follow and clearly written.
Quality and Novelty: Technically solid and proposed approach is novel to the best of my knowledge. The empirical results are ok but not too strong.
Reproducibility: I did not find the code with the paper. I would suggest the authors to release the code for the sake of reproducibility during the review process.

**Strength And Weaknesses:**

Strengths:
- The paper is easy to follow and well-written.
- There has been a lot of research focused on quantization aware training methods in the literature but this paper aims to solve the challenging task of quantized training. It is an important problem to increase training efficiency and this paper makes a good attempt at it.
The proposed method achieves improvements consistently on multiple datasets.

Weaknesses:
- The computational overhead of the approach has been shown to be not significant but it would be useful if the authors put some empirical comparisons on training time of the proposed method vs the baselines.
- In table 1, the results for Mobilenet-V2 are considerably worse than UI8 and DAINT8. Can the authors explain the reason why the proposed method is worse in that setup?


**Summary Of The Paper:**

The paper studies the problem of quantized training where the gradients are quantized as well in addition to quantization aware training methods. The paper builds up on an interesting observation related to the detrimental effect of batch normalisation in quantized training. It is shown that batch norm amplifies the accumulated gradient quantization noise during the backpropagation. Based on this observation, a rectification method is proposed to reduce the negative effect of accumulated gradient quantization noise.

**Summary Of The Review:**

Overall the paper makes a good attempt at the problem of quantized training and the proposed idea seems novel as well as technically solid to me. Though the results are not particularly strong. Still I would recommend weak acceptance at this stage.

---

> ### Author Response · Authors · 2022-11-09
> **Response to R1**
>
> Dear R1,
>
> Thanks a lot for your recognition of our manuscript and your comments and suggestions!
>
> ---
>
> __Empirical Computation Overhead.__
>
> Thanks for your great suggestion! We tested the real-world training time of both the proposed method and the baseline. On ImageNet, when training ResNet-18, the baseline took 43.4h, while our method took 44.6h – only changed by 0.67%. We also got similar results on MobileNet-v2. The details are added to the __Appendix A.5__. Based on these observations, the empirical overhead of total training time isn’t affected by the proposed method, which is in line with our theoretic calculation.
>
> __
>
> __Lower results for Mobilenet-V2__
>
> As explained in Sec. 7.1 §1:
> > despite that we simply use vanilla quantizers on the gradient, while the quantizer designs of the counterparts are heavily engineered, e.g. DAINT8 adopts vector quantization to process error signal in channel-wise manner. We found that MobileNet-V2 on ImageNet is harder to train with 8-bit gradients with vanilla quantizer designs even training with our $\mathcal{L}_\sigma$, ending up with around 1% accuracy drop than SOTA.
>
> As suggested in prior works [HFP8](https://openreview.net/forum?id=HkxIKNSeIH), MobileNet-V2 are more difficult to quantize and more sensitive to quantization noise.  We also suspect that it was because of the use of vanilla quantizers in our experiments that fails to achieve SOTA results in DAINT8 (even for our baseline). Yet, we still observed an improvement using our method from the baseline.
>
> ---
>
> Please let us know if there are any more comments, thank you.
>
> Sincerely,
>
> Authors.

---

### Author Response · Authors · 2022-11-06
**An Early Response**

Dear reviewers and AC,

We wholeheartedly appreciate your insightful comments provided to help us improve our manuscript. Embracing the feedback, we are making several revisions and conducting supplementary experiments, which we hope to deliver soon.

Before the upcoming revision, however, we would like to briefly address some of the concerns which might have come from a slight misunderstanding; we will provide detailed clarifications in the reply to each individual review.

Best regards,
Authors.

---

### Author Response · Authors · 2022-11-09
**Summary of revisions**


Dear Reviewers and ACs,

Thank you again for your time and effort to provide detailed feedback on our manuscript.
To best appreciate your comments, we conducted several supplementary experiments and revised the manuscript.

The updates for listed below.
- Changed the wordings with similar meaning to “backprop”, “back-prop”, etc. to a unified word “backpropagation” (R2).
- Replaced all notations of quantization noise from $\Delta$ to $e(\cdot)$ (R2).
- Removed the color scheme in Fig 3 (R2).
- Added explanations of denotations of $D(\cdot)$, $\mathbf{g}$, etc. to “Key Notations” part in Section 3 (R2, R3).
- Moved the part of Section 5 into “Related Work” (R2).
- Increased the font sizes in Fig 3-6 for readability (R2).
- Added the missing reference for Table 3 (R2).
- Added Assumption 1 before Theorem 1 (R2).
- Corrected typos in Eq 1 (R1) and Corollary 1.1 (R3).
- Revised the explanation of Fig 2 in Section 4 (R2).
- Added additional experiments of empirical computation overhead (R1).
- Added additional experiments of ablation study of $\lambda$ (R3).

The modified parts are marked in __blue__ in the revised draft.

For more details, we provide responses and clarifications to the reviewers’ comments and concerns individually.

Best regards,

Authors.

---

### Comment · Area_Chair_wXtw · 2022-12-08
**AC comment**

### Description

The goal of the work is to improve the training cost of quantized networks. The method considered is to quantize backpropagation. However when the gradients are quantized the errors accumulate during backpropagation. The theoretical analysis of the error amplification by BN is proposed. The method penalizes the variance statistic of BN to make it not too small, which prevents the growth of accumulated noise variance.

The paper addresses an interesting and challenging problem and demonstrates experimental results that appear promising. However, I see several substantial problems which would require at least a major revision to make it to a solid paper.

### The main goal to improve the training speed is not demonstrated

The training with quantized gradients is more stochastic and may converge slower in iterations, in particular it may require a smaller learning rate. For the approach to be practically relevant it is necessary to demonstrate that the speed improvement by quantizing gradient is worth it. It would be necessary to see the training loss vs. clock time for training without quantized gradients (using its optimal learning rate) and training with quantized gradients (using its optimal learning rate). Since the most costly operation for the backward pass is the transposed convolution, it is not clear that quantization of gradient after BN can result in a speed-up: the backprop of convolution receives channel-wise scaled gradients, i.e. this convolution is not benefitting from quantization. The paper does not discuss these aspects at all.

### The method is not sound

1) Any network with BN has weight scale equivariances. It is possible to achieve $\sigma=1$ simply by dividing all weights $w$ of the preceding linear transform by $\sigma$ and  simultaneously multiplying the affine scaling parameter by the same value. The effect on the gradient propagation is obviously identical. In a ReLU network, the division can go as well to the preceding layer because ReLU is 1-homogenous. It is therefore trivial to achieve $L_\sigma = 0$ and the effect on the accumulated error problem is unclear.

2) The assumption that both inputs $\hat x$ and the errors $\delta$ have the same variance is not justified and not verified.
The derivation used in Theorem 1 can be equivalently applied to the error $e$ and to the gradient $g$. Therefore the variance of the accumulated noise encounters the same scale factor as the variance of the gradient. It does not imply that SNR gets any worse. The dominant effect is seen to be just scaling by $1/\sigma^2$ as would be in a layer applying a constant scaling factor. Therefore the analysis does not allow to make any conclusions. In addition, a sequence of BN layers only, without gradient quantization, would obviously not amplify the noise, i.e. decrease SNR "exponentially". So the chain of the quantization and other ops including BN needs to be modelled for the theoretical analysis.

### Novelty / Literature

The method is not entirely novel. The theoretical analysis of this kind is rather common for studying vanishing / exploding problems in NNs, starting with He initialization. More advanced studies exist under the name "mean field theory of NNs", in particular for BN:

Yang et al. (2018) A Mean Field Theory of Batch Normalization

The following basic reference is also missing:

Gupta et al (2015): Deep Learning with Limited Numerical Precision,

I believe considering stochastic rounding of weights, activations and gradients. The submission does not detail which kind of quantization is used, which is an additional clarity issue.

A baseline alternative to BN is not to remove it completely but to apply it as an initialization technique: accumulate $\mu, \sigma$ and then replace it by the test-time equivalent affine layer:

Mishkin (2015): All you need is a good init

Then, regularization techniques on the mean and variance statistics have been proposed as an alternative for BN in binary networks:

Ding et al. (2019) Regularizing Activation Distribution for Training Binarized Deep Networks

### Clarity

Several issues have been pointed by reviewers. Variance is often discussed outside of context of Theorem 1 or BN, e.g. in 7.3. It is not clear what is the probability space, i.e. what is considered random. Fig 1. is not clear and shows apparently training accuracy and not the loss in b). Also in f) the test accuracy show is below that of a random guess for CIFAR-10, apparently something is wrong.

---

### Decision · Program_Chairs · 2023-01-20

**Decision:**

Reject

**Justification For Why Not Higher Score:**

Serious flaws

**Justification For Why Not Lower Score:**

N/A

**Metareview: Summary, Strengths And Weaknesses:**

### Decision

The paper addresses an interesting and challenging problem and demonstrates experimental results that appear promising.
The paper received low or borderline initial scores. The main reason for rejection by reviewer FcHZ seems to be the lack of clarity and rigorousness. The author's rebuttal addresses this to some extent. The reviewer GegD was not convinced by the paper, expecting a more comprehensive theoretical and experimental analysis. The average rating implies rejection. I have elaborated on two issues in detail the "AC comment" below, which are in my opinion serious and also imply rejection.